# Sex-Specific Polygenic Risk Scores and Replication in a Model-Free Analysis of Schizophrenia Data

**DOI:** 10.3390/genes16091080

**Published:** 2025-09-15

**Authors:** Anna Ott, Jurg Ott

**Affiliations:** 1Center of Statistical Genetics, West Orange, NJ 07052, USA; 2Laboratory of Statistical Genetics, Rockefeller University, New York, NY 10065, USA

**Keywords:** polygenic risk score, cross-validation, sex-specific analysis, sex differences, genetic risk, psychiatric genetics, polygenicity

## Abstract

Background/Objectives: While single variants may have only small effects on common heritable traits like schizophrenia, methods for combining such effects over multiple variants have been proposed for more than 30 years. The currently favored approaches are polygenic risk scores. Their main aim is the genetic prediction of phenotypes. Methods: To accommodate the inherent genetic heterogeneity between males and females, we separated them into two independent datasets and in each developed allelic polygenic risk scores. We focused on variants with high predictability rather than high statistical significance and derived a statistical test to assess the significance of results obtained in one sex and replicated in the other sex. Results: As few as 5000 highly predictive variants achieved accuracy exceeding 95% in each of males and females, and only 2.8% and 3.3% of cases and controls were misclassified in females and males, respectively. Conclusions: Our allelic polygenic risk scores are based on individual genotypes rather than summary statistics and produce highly accurate, cross-validated phenotype predictions. Although variants were originally selected as being highly predictive rather than statistically significant, 544 disease-associated variants were shown to be significantly shared between males and females, which represents a replication in an independent dataset.

## 1. Introduction

Schizophrenia is a psychiatric trait affecting approximately 1% of the population and has a heritability of 80% [1]. The genetic contribution to this trait is mainly due to susceptibility variants with common minor alleles (>1% frequency) of small effects, and very few of these variants achieve genome-wide statistical significance [2]. Even a simple polygenic threshold model with 100 polygenes explains empirical segregation ratios (proportions of affected offspring by parental phenotypes) for schizophrenia much better than single-gene inheritance models [3]. Here, we combine single-locus effects over large numbers of risk variants and use resulting sets of highly predictive variants to predict disease phenotypes in a previously published case–control dataset for schizophrenia.

Current genomic approaches only explain around 40% of heritability [2], which speaks to the extreme polygenicity of schizophrenia. Consequently, methods are needed to extract information from the combined effects of large numbers of genetic variants.

Various approaches have been taken to combine single-variant effects over multiple variants. Early efforts include summing maximum LOD scores over different variants [4] and summing variant-specific statistical test results over variants [5,6]. Currently, the favored approach is polygenic risk scores (*PRS*) [7], which, in the simplest case, combine variant-specific effects in the form of weighted averages of risk alleles over a possibly large number of variants. Each study individual will then be assigned a risk score—the larger this score, the higher the probability for an individual to be a “case” (affected with disease) rather than a “control” (unaffected). The main purpose of a *PRS* is genetic prediction of the disease phenotype, which can be performed early in life, well before actual onset of a trait.

Different forms of *PRS* exist, from basic allelic *PRS*, as implemented in the PLINK version 1.9 [8] software [7], to sophisticated model-based approaches [9,10,11,12,13,14]. Some Bayesian methods can accommodate all variants in a study, although they are usually based on summary statistics rather than individual genotypes [15].

While the purpose of our analysis is risk prediction based on genetic variants, we recognize notable sex differences in schizophrenia spectrum disorders. For example, males have an earlier onset and more severe negative symptoms, and females exhibit better verbal abilities and a more favorable prognosis [16].

Several previously published studies have examined sex-specific variations in polygenic scores. For example, in a longitudinal dataset of healthy individuals, researchers found a significant interaction effect between sex and *PRS* on cognitive task performance, and subsequent sex-stratified analyses showed that the *PRS* effect was male-specific [17]. In a similar analysis of non-clinical participants, linear regression analyses did not furnish any associations between *PRS* and subclinical phenotypes, but a male-specific association was found between *PRS* and positive schizotypy [18]. In contrast to these reports, we will directly analyze males and females separately and will not consider an analysis of the whole sample in order not to be led astray by a potential false positive association due to data heterogeneity rather than inherent association. Also, sex-specific effects are sometimes only elucidated after interaction effects with sex in the whole sample have been observed [17]. Furthermore, higher-order interaction effects might not be seen in the total sample but may be discovered as lower-order effects in separate male and female samples.

## 2. Materials and Methods

### 2.1. Dataset

To maximize genetic homogeneity, we wanted to work with a genetic isolate and downloaded from dbGaP the dataset entitled Genetics of Schizophrenia in an Ashkenazi Jewish Case–Control Cohort, comprising 3096 individuals (1044 cases and 2052 controls; 2164 males and 932 females). Of the original 989,972 variants loaded by plink, 7525 were removed for missing call rates exceeding 0.10; 89,597 variants were monomorphic; 1161 variants significantly violated Hardy–Weinberg proportions in controls; and 84,531 variants showed minor allele frequencies less than 0.01, which left 807,158 variants for analysis. Pruning with plink option, --indep 50 5 2, removed 628,076 variants, leading to 179,082 relatively independent variants.

Details about the dataset are available from the dbGaP website (see Data Availability Statement), from which we quote the following study history:

Inclusion criteria for all subjects: Self-identified 4 grandparents of Ashkenazi Jewish heritage. Inclusion criteria for cases: Hospitalized inpatients meeting DSM-IV criteria for schizophrenia or schizoaffective disorder.

Exclusion criteria for cases: Subjects diagnosed with at least one of the following disorders: Psychotic disorder due to a general medical condition, substance-induced psychotic disorder, or any Cluster A (schizotypal, schizoid, or paranoid) personality disorder. Exclusion criteria for controls: Report of any chronic disease or taking any medication at the time of blood draw.

### 2.2. Sex-Specific Heterogeneity

Many traits manifest differently in males and females. Thus, one of the main biological covariates affecting disease is sex. In logistic regression analysis, sex effects may be accommodated with a binary dummy variable. Its effect is simply to allow the disease prevalence to be different in males and females. However, in many situations, sex effects vary depending on the genetic variants. Thus, to accommodate variant effects specific to one sex, we separated males and females into two independent datasets, each with the same 179,082 variants, and proceeded to analyze them separately. This also allows us to corroborate a finding in one sex by showing its occurrence in the other sex as an independent replication. Separate analyses of males and females were uncommon until just a few years ago, but have now become more accepted. Recent examples are a significant association of polygenic risk scores with heart disease in females but not in males [19] and a sex-specific association between schizophrenia polygenic risk and schizophrenia-related traits [18].

### 2.3. Polygenic Risk Scores

An allelic polygenic risk score is a weighted average of minor alleles over a possibly large number *N* of variants. Equation (1) shows the formula for the *PRS* of the *j*th individual as implemented in plink, disregarding potentially missing observations [7], where *OR*_i_ is the allelic odds ratio of the *i*th variant, and *g*_ij_ = 0, 1, or 2 represents the number of minor alleles at the *i*th variant in the *j*th individual.(1)$${PRS}_{j}=\sum_{i=1}^{N}\log\left({OR}_{i}\right)g_{ij}/(2N)$$

Bayesian approaches to *PRS* construction, often based on summary statistics [12], can accommodate large numbers of all variants in a study. Other approaches initially include in a *PRS* the most significant variants, followed, if necessary, by additional variants despite them being non-significant [7,20]. However, significant variants tend to be poor predictors, and variants with excellent predictive ability are generally not significant [21,22,23]. Thus, we initially recruit the most predictive variants, that is, those with the largest odds ratios (see next section), whether or not they are statistically significant. Also, for optimal accuracy, we do not impose models on our data and work directly with individual genotypes.

### 2.4. Prediction

For a given number of variants included in a *PRS*, each individual receives a score, which tends to be higher for cases than controls, although the two score distributions generally overlap to some degree. We predict an individual to be a case if their score exceeds the 95th percentile of control scores; otherwise, the individual is predicted to be a control. This approach is essentially model-free and is analogous to defining elevated lipid levels as those above the 95th percentile of population levels [24]. Thus, we are working with the following decision table, where *a*, *b*, *c*, and *d* stand for numbers of individuals [25]. For example, *a* = number of cases predicted to be cases.

The quality of prediction may be measured by the positive predictive value, *PPV* = *a*/(*a* + *c*), which is the proportion of known cases among individuals predicted to be cases. Analogously, the negative predictive value, *NPV* = *d*/(*b* + *d*), is the proportion of controls among individuals predicted to be controls. The odds ratio, *OR* = *a* × *d*/(*b* × *c*), may be written as *PPV* × *NPV*/[(1 − *PPV*) × (1 − *NPV*)] and is seen to be increasing with *PPV* and *NPV*. Finally, the prediction accuracy is defined as *ACC* = (*a* + *c*)/(*a* + *b* + *c* + *d*).

In statistical testing, a table similar to Table 1 is used, but the rows refer to the alternative hypothesis (instead of case) and null hypothesis (instead of control), and the two columns stand for a test to be significant or not. Power of the test is then measured by the sensitivity or true positive rate, *TPR* = *a*/(*a* + *b*), while the *p*-value or false positive rate is given by *FPR* = *c*/(*c* + *d*), where 1 − *FPR* is also known as specificity. Often, an *ROC* curve [25] is constructed with *TPR* as the ordinate versus *FPR* as the abscissa, and the area under this curve, *AUC* ≥ 0.5, serves as a measure of overall power. It has become common to construct *AUC* also for prediction tables like Table 1, but we prefer *ACC* over *AUC* as a measure of predictive ability because *ACC* has an easy probabilistic interpretation as the proportion of correct predictions and refers to classification of both cases and controls, while the AUC focuses on cases.

To determine the minimum number of variants needed in a *PRS* to achieve *ACC* ≥ 0.95, we construct a *PRS* for each of a sequence of numbers of variants, *N* = 5, 10, …, 50,000, where *N* refers to the number of variants with the largest allelic *OR*. For given *N*, we find *ACC* for males and females each, and the resulting graphs will demonstrate the number of variants needed to achieve *ACC* ≥ 0.95 in each of males and females.

### 2.5. Replication

If a method works well in a given dataset, it is important to replicate its performance in another dataset. A well-known internal replication method is cross-validation, where prediction rules are developed in a proportion *q* of the data and applied to the remaining 1 − *q* of individuals [26]. A common choice is ten-fold cross-validation, *q* = 0.90, where the proportion *q* is selected randomly, and this selection and verification are repeated many times. However, we prefer the Leave-one-out method [26], *q* = (*K* − 1)/*K*, where one of the *K* individuals is removed from the data, which is then used to develop classification rules from scratch and apply them to that single outsourced individual. This procedure is carried out for each individual and does not involve randomness. All results shown in this analysis have been cross-validated in this manner.

While cross-validation effectively develops and applies classification rules in different datasets, it may be more convincing to actually have two independent datasets at hand. We take advantage of our sex-specific analyses by replicating in one sex what we have obtained in the other sex. As described in detail below, we developed software, Replic2 version 19 February 2025, to assess whether such a replication is statistically significant. Both our implementation of *PRS* and Replic2 are available at https://github.com/jurgott. We have not made use of artificial intelligence in our work.

## 3. Results and Discussion

### 3.1. Variant by Variant Analysis

Based on 100,000 permutations of phenotypes in PLINK version 1.9, an initial single-variant case–control analysis showed eight variants in each sex to be significant (*p* < 0.05, corrected for multiple testing), three of which were the same in males and females. Of the total of 13 significant variants, only three are located in genes, with these genes being OR2L13 and LARGE1 in males and MGAT4C in females. In a digenic analysis of the same dataset, these genes were previously shown to form networks of genes associated with schizophrenia [27]. The variants identifying the three genes are all intronic transcript variants.

### 3.2. Polygenic Risk Scores

Next, we constructed *PRS* for females, males, and both sexes combined (Figure 1). The number of variants required for prediction accuracy (*ACC*) to exceed 0.95 is 1000, 5000, and 10,000 in *F*, *M*, and *F* + *M*, respectively. Even though there are more than twice as many males as females, it takes fewer best variants in females than males for *ACC* to reach 0.95, which may be explained by a higher trait heritability in females than males. Also, the combined dataset comprises more individuals than either males or females, yet it requires the most variants for *ACC* to exceed 0.95. This “anomaly” presumably is a reflection of heterogeneity between males and females. Nonetheless, as Figure 1 shows, all three datasets eventually reach a plateau for ACC. It appears that a higher number of variants in a *PRS* can compensate for the degrading effects of heterogeneity, which may be the reason for the generally huge number of variants used for constructing *PRS*.

Actual prediction results are shown in Table 2, which lists, for a selected number of best variants, numbers (*n*) and proportions (%) of misclassified individuals. Females exhibited the smallest misclassification rate of 2.8%—only 26 out of the 932 females were misclassified in a *PRS* with the best 2000 variants. In fact, all 384 cases were correctly predicted to be cases, but of the 548 controls, 26 were predicted to be cases. This highly accurate result demonstrates the power of working directly with genotypes. In males, the *PRS* with 5000 best variants predicted all but one of the cases correctly, but it misclassified 71 of the 1504 controls as cases. The *F* + *M* dataset misclassified 105 individuals, more than the number misclassified by females (*n* = 26) and males (*n* = 72) combined. Clearly, it is advantageous to separate males and females and analyze them separately unless no heterogeneity exists between the two sexes.

As Table 2 shows, optimal prediction accuracy (minimum misclassification rate) with the smallest possible number of variants included in a *PRS* may be achieved with as few as 5000 variants.

For comparisons with other publications, we also computed empirical *ROC* curves [25] and obtained resulting *AUC* values based on *TPR* and *FPR* ratios for values of variants ranging from 5 through 50,000. Proper *ROC* curves meet at least two conditions: (1) *FPR* ≥ *TPR*, and (2) *TPR* ratios are non-decreasing. Thus, we smoothed the original ragged curves by repeated applications of overlapping 3-point moving averages until the desired conditions were met. Results are shown in Figure 2 and demonstrate appreciable *AUC* values, which are in line with the excellent *ACC* values obtained above.

### 3.3. Replication

To determine how many highly predictive variants were shared by males and females, we selected 5000 variants with the largest *OR*s from the 179,082 variants in each sex. It turned out that *N*_s_ = 544 of these 5000 variants each were the same. To find out whether this amount of sharing is unusually high, we developed a replication test, implemented in the Replic2 software. It works by simulating the null hypothesis of randomly selecting a subset of variants in each of males and females and recording the resulting number of variants shared just by chance. This procedure is repeated 100,000 times, and the empirical significance level, *p*, is estimated by the proportion of 100,000 randomly sampled variants as large or larger than the actually observed number of shared variants. Sharing 544 variants turned out to be unusually high, *p* < 0.00001. Thus, these 544 variants were significantly replicated. The median number of variants shared by chance in this case is only 139.5.

To whittle these 544 variants down to a manageable number, we selected variants with an allelic odds ratio (*OR* ≥ 2), which resulted in 15 variants shared by males and females, and 10 of these variants resided in genes (Table 3). None of these variants/genes were significantly associated with schizophrenia on their own, but their disease association in one sex was significantly replicated in the other sex.

Several of the genes listed in Table 3 have previously been implicated in the etiology of schizophrenia:The CUL9 gene was recently shown to be one of five genes causally related to schizophrenia [28].In older literature, the PTPRZ1 gene was genetically associated with schizophrenia, and PTPRZ1-transgenic mice exhibited molecular and cellular changes implicated in the pathogenesis of schizophrenia [29].In a bivariate meta-analysis, MED19 and other genes were found to be associated with schizophrenia [30].Darier disease, an autosomal dominant skin disorder, is caused by mutations in the ATP2A2 gene [31]. In a population-based study, relatives of individuals with Darier disease had a significantly higher risk of having bipolar disorder than relatives of matched individuals from the general population, suggesting that genetic variability within the ATP2A2 gene also confers susceptibility for bipolar disorder [31], but no such association was found with schizophrenia. Our results now strongly demonstrate that the ATP2A2 gene is in fact associated with schizophrenia.

Regarding the variants and genes in Table 3 not previously implicated in being associated with schizophrenia, we strongly feel that these significant associations deserve a close look by specialists in this field.

### 3.4. Limitations and Future Research Directions

The dataset shows a near 1:2 ratio of cases to controls and female to male participants. Such imbalances are known to reduce statistical power, and a recent simulation study has shown exactly that [32]. They do not seem to have other negative effects.

Another “limitation” is that controls have been chosen to be from the same ethnic and population background as the cases. While such a choice has been necessary for optimal power and unbiased results, a distant aim for future research is the ability to predict the disease phenotype of a given single individual based on a generally available database of control individuals, where controls are not necessarily from the same ethnic background as single cases. Such general individualized genetic predictions are not currently possible to any degree of accuracy, and much modeling and statistical research will be required to achieve such often-stated goals.

## 4. Conclusions

Our model-free approach to constructing *PRS* furnishes highly accurate predictions with relatively modest numbers of variants. One reason for this excellent result might be that we work directly with genotypes rather than summary statistics. Also, sex-specific analyses avoid the potentially negative effects of heterogeneity between males and females. On the other hand, more heterogeneous datasets are expected to furnish less predictive results.

Carrying out separate analyses for males and females is still not often carried out, although there have been an increasing number of such reports [33,34,35,36,37]. Of course, in model-based analyses, it is more economical to allow for sex effects through a binary dummy variable, but such a variable can only capture main effects of sex on disease prevalence, which may be all that is needed, although more intricate differences between males and females cannot be addressed in this manner.

Quality of analyses involving *PRS* is often judged by the *AUC*. However, this measure seems more appropriate for the power of a statistical test than the performance of prediction. Thus, we prefer accuracy, *ACC*, to assess prediction performance. In addition to being directly related to predicted and known phenotypes, *ACC* handles cases and controls in a symmetric manner, which is not the case for *AUC*.

## Figures and Tables

**Figure 1 genes-16-01080-f001:**
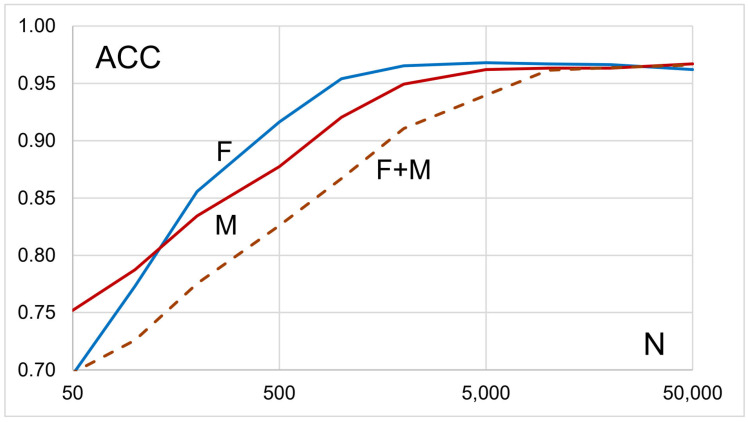
Polygenic risk scores for different numbers *N* of best variants (largest *OR*s) in females (*F*), males (*M*), and males and females combined (*F* + *M*).

**Figure 2 genes-16-01080-f002:**
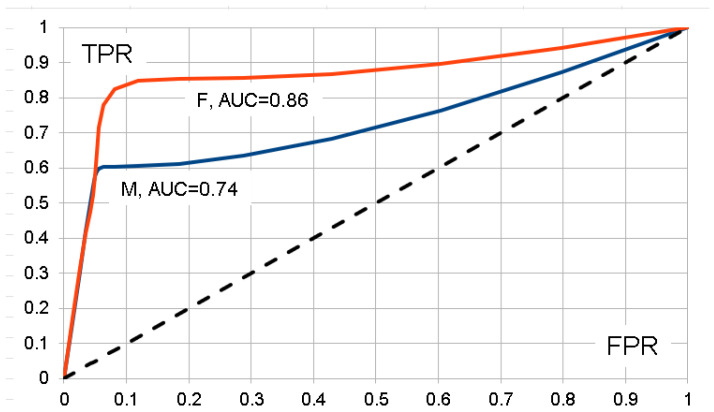
Plot of true positive rates (*TPR*) versus false positive rates (*FPR*) for males (*M*) and females (*F*), and resulting areas under the curve, *AUC*. The dotted line refers to *TPR* = *FPR*, *AUC* = 0.5.

**Table 1 genes-16-01080-t001:** Phenotype predictions versus known phenotypes. Numbers of individuals, *a* … *d*, for example, *a* = number of cases predicted to be cases.

Known Phenotype	Prediction of “Case”	Prediction of “Control”
case	*a*	*b*
control	*c*	*d*

**Table 2 genes-16-01080-t002:** For *N* variants with the largest odds ratios, the number (*n*) and proportion (%) of individuals misclassified in females (*F*), males (*M*), and males and females combined (*F* + *M*). Minimum misclassification rates are shown in bold and underlined.

*N*	*n*, *F*	%, *F*	*n*, *M*	%, *M*	*n*, *F* + *M*	%, *F* + *M*
100	207	22.2%	446	20.6%	851	27.5%
200	131	14.1%	371	17.1%	702	22.7%
500	65	7.0%	260	12.0%	534	17.2%
1000	37	4.0%	167	7.7%	382	12.3%
2000	26	** 2.8% **	91	4.2%	323	10.4%
5000	34	3.6%	72	** 3.3% **	126	4.1%
10,000	30	3.2%	84	3.9%	110	3.6%
20,000	29	3.1%	83	3.8%	121	3.9%
50,000	35	3.8%	71	3.3%	105	** 3.4% **

**Table 3 genes-16-01080-t003:** Ten variants significantly shared by males and females and residing in genes; *chr* = chromosome, *bp* = base-pair position according to assembly GRCh38.

*chr*	*Variant*	bp	*Gene*	Variant Type
5	rs2250599	14,861,105	*ANKH*	intron, genic upstream transcript
6	rs16896326	43,185,671	*CUL9*	intron, genic upstream transcript
6	rs11962528	67,901,631	*LOC105377845*	intron
7	rs1196509	121,975,547	*PTPRZ1*	intron
10	rs978554	99,742,899	*CUTC*	intron
11	rs10896638	57,703,469	*MED19*	500B downstream transcript
12	rs9540	110,351,050	*ATP2A2*	3’UTR, benign
14	rs2293792	73,249,825	*PAPLN*	intron, genic upstream transcript
15	rs16948431	64,957,012	*ANKDD1A*	intron
17	rs7209186	69,327,056	*ABCA5*	5’UTR, genic upstream transcript

## Data Availability

The schizophrenia case–control dataset used in this study is available from https://www.ncbi.nlm.nih.gov/projects/gap/cgi-bin/study.cgi?study_id=phs000448.v1.p1 (accessed 25 08 2022). The following software packages were used in our analysis and are freely available: L1outPRS, https://github.com/jurgott/L1PRS/; Plink, https://www.cog-genomics.org/plink/; Replic2, https://github.com/jurgott/Replic2.

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
