# Peer review of "Sex-Specific Polygenic Risk Scores and Replication in a Model-Free Analysis of Schizophrenia Data"

_genes, 2025, doi:10.3390/genes16091080_

Round 1

Reviewer 1 Report

Comments and Suggestions for Authors

Review Report

Manuscript title: Sex-specific polygenic risk scores and replication in a model-free analysis of schizophrenia data

Summary:

The submitted manuscript “Sex-specific polygenic risk scores and replication in a model-free analysis of schizophrenia data” is an original article aimed to develop sex-specific allelic polygenic risk scores (PRS) to predict schizophrenia by focusing on highly predictive genetic variants rather than just statistically significant ones. By analyzing males and females separately, the method achieved over 95% prediction accuracy in both groups, with low misclassification rates. A statistical test confirmed that 544 disease-associated variants were significantly shared between sexes, indicating robust replication. The approach uses individual genotype data and offers highly accurate, cross-validated predictions of genetic risk.

General concept comments:

The manuscript is clear, relevant for the field and presented in a well-structured manner. It sounds scientifically and the manuscript meets the criteria for an original article in terms of structure and substance; it is correctly presented according to the requirements of the journal. The cited references are mostly recent publications (within the last 5 years) and relevant without including excessive number of self-citations. The quality of the writing is highly satisfactory, as well as the figures and tables, which properly show the data and are easy to understand. Data availability statement is adequate.

The sections are well conducted and structured and support the interpretation of the presented literature data. The content of the study would be of great interest to the scientific audience, because of its strengths to perform a model-free approach to constructing polygenic risk scores (PRS) that achieve high prediction accuracy using relatively few genetic variants, partly because it uses individual genotype data rather than summary statistics. It becomes clear that performing sex-specific analyses improves accuracy by avoiding heterogeneity between males and females - an aspect often overlooked in favor of simpler model-based methods using binary sex variables. The authors argue that prediction performance is better assessed using accuracy (ACC) rather than AUC, as ACC directly relates to phenotype prediction and treats cases and controls symmetrically.

However, the following revisions are necessary to improve the clarity, precision, and presentation of the manuscript:

Recommendations:

  1. Discussion Section – Structure and Content:

The current Discussion section is notably brief and closely resembles a Conclusion. It is recommended that the authors expand this section to include a more in-depth interpretation of the findings, their implications, and how they relate to existing literature. If the authors intend to merge the Results and Discussion sections, the current section should be retitled appropriately (e.g., “Results and Discussion”), and a separate, clearly defined Conclusion should be provided, no matter this section is not mandatory according to the requirements of the journal.

  1. Limitations of the Study:

The manuscript lacks discussion of the study’s limitations, which is essential for contextualizing the findings. For example, the type of schizophrenia studied is not specified. Clarifying whether specific subtypes were included (e.g., paranoid, disorganized, catatonic) could impact on the interpretation and generalizability of the results. Addressing such limitations would improve the scientific rigor and transparency of the work.

Rating the Manuscript:

Novelty: The question is original and well-defined.

Scope: The manuscript addresses the aims and scope of the journal.

Significance: The article presents a model-free approach to constructing polygenic risk scores (PRS) that achieve high prediction accuracy using relatively few genetic variants, partly because it uses individual genotype data rather than summary statistics.

Quality and Scientific Soundness: The article is written in an appropriate way and meets the standards of an original article.

Interest to the Readers: The topic is highly relevant to the field, and it would be of interest to the scientific audience.

Overall Merit: This study offers a valuable contribution to psychiatric genetics by examining sex-specific polygenic risk scores (PRS) in schizophrenia, using a model-free replication approach. The methodology is innovative and helps address known clinical sex differences in the disorder. The use of independent cohorts strengthens the validity of the findings, and the model-free design reduces reliance on assumptions. However, some revisions are necessary to improve the clarity and presentation of the manuscript. Despite this, the work opens important avenues for more personalized and biologically informed risk prediction in schizophrenia research.

English Level: The English language is appropriate and understandable.

Author Response

Note: The updated manuscript consists of the original manuscript with changes highlighted by the Track Changes feature under Review (Word). We thank reviewers for their thoughtful comments and excellent suggestions, which greatly improved our manuscript.

Reviewer 1

Comment 1: The current Discussion section is notably brief and closely resembles a Conclusion. It is recommended that the authors expand this section to include a more in-depth interpretation of the findings, their implications, and how they relate to existing literature. If the authors intend to merge the Results and Discussion sections, the current section should be retitled appropriately (e.g., “Results and Discussion”), and a separate, clearly defined Conclusion should be provided, no matter this section is not mandatory according to the requirements of the journal.

Response 1: We agree with the reviewer and provide the following response: We merge the Results and the Discussion sections into a “Results and Discussion” section and rename the current Discussion section “Conclusion”. We also expand the Results and Discussion section and outline these changes in our response to Reviewer 2, whose recommendations are rather specific.

Comment 2: The manuscript lacks discussion of the study’s limitations, which is essential for contextualizing the findings. For example, the type of schizophrenia studied is not specified. Clarifying whether specific subtypes were included (e.g., paranoid, disorganized, catatonic) could impact on the interpretation and generalizability of the results. Addressing such limitations would improve the scientific rigor and transparency of the work.

Response 2: Analogous and more specific comments were made by Reviewer 2 and we agree with these comments. We added a Limitations subsection to the Results and Discussion section and otherwise respond to Reviewer 2 in detail.

Reviewer 2 Report

Comments and Suggestions for Authors

Reviewer Comments

The manuscript addresses an important and timely topic, exploring the role of gender in polygenic risk scores within a schizophrenia-related subject database. The use of online tools for statistical risk analysis is appropriate and relevant. However, several points need to be addressed to enhance the clarity, rigor, and overall impact of the work.

  1. Novelty and Gap Justification
    Several previously published studies (e.g., https://doi.org/10.1038/s41398-021-01649-4, https://doi.org/10.1016/j.pnpbp.2024.111161, among others) have examined sex-specific variations in polygenic scores. The authors should explicitly highlight how the present study addresses gaps not covered by prior work. This can be incorporated into the concluding section of the Introduction to clearly justify the novelty and necessity of the current investigation.
  2. Expanded Background and Context
    While the manuscript provides background on schizophrenia, case–control structure, polygenic risk scores, and statistical modeling, it lacks depth in several critical areas. The Introduction should be expanded to include:
    • A clearer articulation of the core hypothesis.
    • Relevant aspects of schizophrenia pathophysiology.
    • Epidemiological data on sex differences in schizophrenia.
      This will better contextualize the study for readers who may not be deeply familiar with these concepts.
  3. Polygenicity and Heritability
    Please elaborate on the heritability estimates of schizophrenia, linking them to the concept of polygenicity. This will strengthen the theoretical framework for the use of polygenic risk scores in this context.
  4. Dataset Composition and Potential Bias
    The dataset shows a near 1:2 ratio of cases to controls and female to male participants. The authors should discuss whether this imbalance could influence the results or introduce bias. If so, please elaborate on potential mitigation strategies (e.g., statistical adjustments, stratification, or matched sampling) that could be employed.
  5. Preclinical Evidence and Translational Potential
    It would be beneficial to discuss whether preclinical studies have provided evidence for sex-specific genetic influences in schizophrenia. Additionally, comment on the potential for translating such findings into clinical applications.
  6. Expanded Discussion, Limitations, and Conclusion
    The Discussion is currently too brief. It should:
    • Relate the findings to existing preclinical and clinical studies (with appropriate references).
    • Address potential therapeutic implications and future research directions.
    • Include a clear Limitations subsection.
    • Add a distinct Conclusion section summarizing the key contributions and implications of the study.
  7. Keywords
    Consider adding more keywords that capture the core concepts of the manuscript (e.g., “sex differences,” “genetic risk,” “psychiatric genetics,” “polygenicity,” “sex-specific analysis”) to improve discoverability in academic searches.

By addressing these points, the manuscript will be more comprehensive, contextually grounded, and impactful for the field.

Author Response

Comment 1: Several previously published studies (e.g., https://doi.org/10.1038/s41398-021-01649-4, https://doi.org/10.1016/j.pnpbp.2024.111161, among others) have examined sex-specific variations in polygenic scores. The authors should explicitly highlight how the present study addresses gaps not covered by prior work. This can be incorporated into the concluding section of the Introduction to clearly justify the novelty and necessity of the current investigation.

Response 1: We agree with this comment and added a paragraph at the end of the Introduction section.

Comment 2: While the manuscript provides background on schizophrenia, case–control structure, polygenic risk scores, and statistical modeling, it lacks depth in several critical areas. The Introduction should be expanded to include:

    • A clearer articulation of the core hypothesis.
    • Relevant aspects of schizophrenia pathophysiology.
    • Epidemiological data on sex differences in schizophrenia.

This will better contextualize the study for readers who may not be deeply familiar with these concepts.

Response 2: We agree with this comment and made the following changes: Early on in the introduction, we define the hypothesis we want to explore. We also quote epidemiological data on sex differences in schizophrenia. Finally, we added some paragraphs to the 2.1 Dataset section, indicating inclusion and exclusion criteria for cases and controls.

Comment 3: Please elaborate on the heritability estimates of schizophrenia, linking them to the concept of polygenicity. This will strengthen the theoretical framework for the use of polygenic risk scores in this context.

Response 3: We added a small paragraph relating heritability to polygenicity, quoting Owen et al (2023).

Comment 4: The dataset shows a near 1:2 ratio of cases to controls and female to male participants. The authors should discuss whether this imbalance could influence the results or introduce bias. If so, please elaborate on potential mitigation strategies (e.g., statistical adjustments, stratification, or matched sampling) that could be employed.

Response 4: In a newly added paragraph, 3.4 Limitations, we quote a recent study showing optimal power for 1:1 ratios of cases and controls (and sexes).

Comment 5: It would be beneficial to discuss whether preclinical studies have provided evidence for sex-specific genetic influences in schizophrenia. Additionally, comment on the potential for translating such findings into clinical applications.

Comment 6: The Discussion is currently too brief. It should:

    • Relate the findings to existing preclinical and clinical studies (with appropriate references).
    • Address potential therapeutic implications and future research directions.
    • Include a clear Limitations subsection.
    • Add a distinct Conclusion section summarizing the key contributions and implications of the study.

Responses 5 and 6:

            Preclinical and clinical studies – above, we briefly mentioned epidemiological data on sex differences in schizophrenia, but beyond that, as non-clinicians, we’d rather not add more comments.

            Limitations, Future research directions – we added a subsection, 3.4 Limitations and future research directions.

            Conclusion section – done.

Comment 7: Consider adding more keywords that capture the core concepts of the manuscript (e.g., “sex differences,” “genetic risk,” “psychiatric genetics,” “polygenicity,” “sex-specific analysis”) to improve discoverability in academic searches.

Response 7: Done